# Locality in the Promoted Sustainability Practices of Michelin-Starred Restaurants

Yuying Huang [1,*] and C. Michael Hall [1,2,3,4,5,6,*]

1   Department of Management, Marketing and Tourism, University of Canterbury, Christchurch 8041, New Zealand
2   Geography Research Unit, University of Oulu, 90014 Oulu, Finland
3   Department of Service Management and Service Studies, Lund University, 25108 Helsingborg, Sweden
4   Department of Organisation and Entrepreneurship, School of Business and Economics, Linnaeus University, 39234 Kalmar, Sweden
5   Centre for Research and Innovation in Tourism, Taylor's University, Subang Jaya 47500, Malaysia
6   College of Hospitality and Tourism Management, Kyung Hee University, Seoul 02447, Republic of Korea
*   Correspondence: yuying.huang@pg.canterbury.ac.nz (Y.H.); michael.hall@canterbury.ac.nz (C.M.H.)

**Abstract:** Sustainable practices are increasingly promoted in the restaurant industry. One significant aspect of sustainability in restaurants is the use of local supply chains, especially for food, which also serve as a means for restaurants to promote freshness of produce, sourcing, and quality. Considering the prevalence of locality in menu marketing, this study aims to explore the relationships between sustainability and locality at fine-dining restaurants. Michelin-starred restaurants are significant influencers in the restaurant industry, as well as food fashions overall, and may therefore serve to promote sustainability practices. This study examines the sustainability of 135 Michelin three-star restaurants by conducting website content analysis. By identifying restaurants' sustainable practices during the processes of procurement, preparation, and presentation and analysing the official websites of 135 Michelin three-star restaurants, this study finds that although all sustainable practices are mentioned by less than half of the reviewed websites, most practices could be interpreted as being embedded in their locality, especially local food and restaurant history. This study suggests that promoting locality could therefore help sustain sustainability in the fine-dining restaurant industry. Although this study is limited to the website content of official websites for Michelin three-star restaurants, it provides potentially valuable insights on the promotion of sustainable restaurant practices.

**Keywords:** sustainable restaurants; terroir restaurants; fine dining; Michelin Guide; food system; foodservice business

## 1. Introduction

Sustainability affects every field of human endeavour, including the culinary system and restaurant industry [1,2]. Previous research has indicated five basic principles of sustainability: (1) the idea of holistic planning and strategy-making that links economic, environmental, and social concerns; (2) the importance of preserving essential ecological processes; (3) the need to protect both biodiversity and human heritage; (4) the need for development to occur in such a way that productivity can be sustained over the long term for future generations; and (5) the goal of achieving a better balance of fairness and opportunity (equity) between and within nations [3] (pp. 20). Several studies have provided insight into the economic, social, and environmental dimensions of sustainability in the restaurant industry [4,5], with criteria developed to evaluate a restaurant's sustainability, such as restaurant and food service life cycle assessment [6], and Green Restaurants ASSessment (GRASS) [7]. A number of organisations have also launched accreditation schemes and certifications to distinguish sustainable restaurants, such as the Michelin Green Star [8],

Food Made Good by the Sustainable Restaurant Association [9], and Dine Green by the Green Restaurant Association [10].

However, restaurant sustainability practices might not be recognized by customers, thereby affecting their dining decisions in and selection of green or sustainable restaurants [11]. Some customers might even doubt sustainable claims due to concerns over greenwashing, e.g., misleading or empty environmental claims [12]. In the restaurant context, if restaurants only take initiatives visible to customers, such as using the word 'green' as a marketing ploy, but ignore initiatives that can actually do good, their efforts may be perceived as greenwashing and destroy trust in restaurant sustainability initiatives [11,13]. In addition, some restaurant sustainability initiatives may be relatively expensive or have up-front costs, such as changing energy-saving appliances [14], purchasing ethical food (organic food, sustainable seafood) [15], and obtaining green certifications [16]. Some restaurants might give up maintaining these practices in the long term if they have some financial difficulties, especially following crises like COVID-19 that affect their capacity to operate [17]. In light of these issues, the current study advocates that restaurants should search for appropriate and cost-effective ways to present sustainability practices.

Sustainability in the culinary system is usually examined with respect to the three dimensions of environmental, social, and economic sustainability [4]. Environmental benefits include low carbon emissions in terms of transportation, packaging, and storage [18]; food security and biodiversity conservation [19]; and the adoption of environmentally-friendly growing practices (i.e., the use of hedges, organic farming, and traditional species grown) [20]. Social benefits include the conservation of food, culture, history, and geography [21]; the development of new forms of tourism (i.e., rural tourism, food tourism) [22]; and the enhancement of local community ties [23]. Economic benefits include the economic viability of enterprises or food products, diversification (sources of farm income), food events and festivals, as well as destination attractiveness [24]. Within these perspectives, sustainability practises in a restaurant may not only benefit the restaurant itself, but also positively influence the local food system and the local community. Therefore, localism has been advocated for in the restaurant industry [1,2] with, for example, scholars identifying significant relationships between locality and sustainability in destination restaurant research [25].

Localism is usually contrasted with globalism in the food system [24,26]. The global industrialised food system and its key actors, such as supermarkets and fast-food chains, have inspired a range of responses to its perceived homogenising effects, ranging from environmental and ethical responses to the creation of new forms of culinary tradition and the presentation of gastronomic authenticity [27]. Globalisation is often regarded as a process that makes geography less relevant for social and cultural arrangements, which have impaired the diversity and quality of the local food culture [28]. In contrast to globality, locality has become an increasingly prevalent food trend in recent years which could be regarded as a form of local cultural capital that strengthens sense of place, place and product differentiation, and place branding in a globalised world [24,29,30]. For example, one of the world's leading restaurants, Noma in Denmark with Chef René Redzepi, is a strong promoter of regional gastronomy and "the New Nordic movement", that stresses a set of principles including locality, sustainability, and respect for the natural world in a Nordic place context [31]. Previous studies have identified restaurants that promote locality as "terroir restaurants" and noted the relationship between sustainability and locality [21,32,33]. Considering the significant role of restaurants in the culinary industry and the food system in general [1], it is therefore timely to investigate the empirical reality of sustainable practices at restaurants, and how these practices contribute to locality from the restaurant perspective.

This study aims to understand the relationships between sustainability and locality in Michelin-starred restaurants (MSRs) because of their high profile in the foodservice industry and the culinary system [34]. The research questions of this study are: (1) the extent to which sustainable practices are promoted as part of MSRs' offerings; and (2) how these practices

reflect the locality. Nine sustainable practices in the processes of procurement (i.e., local food, farm-to-table activity, food foraging), preparation (i.e., efficient restaurant, sustainable menu, minimising waste), and presentation (i.e., restaurant's history, community outreach, cooking school) are identified based on the literature and then analysed within the sample of Michelin three-star restaurants' official website content. The discussion aims to clarify the characteristic of the high-profile terroir restaurant and their sustainable practices in terms of presenting locality. This study concludes by noting the importance of locality and its contribution to sustaining sustainability for the restaurant industry and the culinary system.

## 2. Literature Review

### 2.1. Michelin-Starred Restaurants and Sustainability

The Michelin Guide is regarded as a tastemaker of contemporary food and restaurant culture, wielding both symbolic and material power in the global restaurant industry [35,36]. MSRs are usually positioned as fine-dining restaurants which offer high-quality food, innovative menus beyond the norm, pricing that is expensive compared to casual restaurants, and social status and self-expression values when consumed [37]. Although customers usually do not visit those higher pricing MSRs as regularly as casual restaurants [38], MSRs still have profound impacts at individual, sector, and societal levels because of their prestige [39]. MSRs could also lead food trends and promote culinary-related initiatives in the food system due to the high profile of Michelin Stars in food media and the wider restaurant industry [2]. For example, during the pandemic crisis, many MSRs took initiatives centred on consumer and food well-being, implemented socially responsible business practices to support foodservice actors, and launched philanthropic activities targeting community well-being [40]. By doing these, MSRs help tackle social issues, fulfil their social responsibility and sustainability goals, and improve collective and individual well-being [2].

In recent decades, discussion and innovation about sustainability have become increasingly prevalent in the foodservice industry [2]. Previous studies have examined the sustainable practices implemented by restaurants that have critical influences at individual, sector, and societal levels. For individual customers, some restaurants design sustainable menus by marking each dish with a carbon label, local farmer label, and rankings of general environmental impacts, which could further encourage to customers order sustainable-labelled food [41–43]. For the foodservice sector, the procurement of local food is beneficial to sustaining the local food system, culinary culture, and biodiversity [20,44,45]. Restaurants rooted in their regions, relying on local products, and representing place can be regarded as a tool for local development, providing opportunities for place promotion and reinforcing the local economy and supply chain [2,24,25]. However, few studies explored sustainability in fine-dining restaurants, and whether luxury food and sustainability are compatible or not remains unknown [46]. Some luxury dining experiences have even been thought to be incompatible with sustainability, given that traditionally such fine-dining restaurants were often known for their focus on abundance, use of rare species, disregard for seasonality, and fatty dishes, which run counter to many notions of ethical food consumption and sustainability [47,48]. Therefore, this study aims to examine sustainable practices conducted by MSRs in the terms of procurement, preparation, and presentation [2].

### 2.2. Sustainable Procurement

The use of high-quality ingredients is one of the criteria used to award Michelin stars [49]. Fine-dining restaurants, especially MSRs, apply two strategies to foodstuffs by "making it deluxe" and "making it pure" [50,51]. The former, "making it deluxe", demonstrates the exclusiveness of the ingredients, such as white Alba truffle, Almas caviar, and Kobe-style Wagyu beef. These foodstuffs are recognised as luxury ingredients and production is often limited to specific areas or varieties of food and drink, which require

the restaurants to establish their global food supply chain to obtain them [52]. The latter "making it pure" focuses on localness and ethical soundness that could be represented in the local food system [53]. The growth of localism has been discussed in foodservice research [24,26,54]. Scholars have reviewed two decades of research on local food systems (LFS) and generalised eight benefits of LFS within the perspectives of consumers, farmers, community, and environment: LFS can increase consumers' access to fresh and healthy food; consumers are willing to pay more for local over non-local food; farmers have a great sense of social recognition in LFS; participation in LFS economically benefits farmers; LFS increase local community ties; LFS benefit the local economy; LFS foster environmentally-friendly production practices; and LFS help mitigate climate change [20]. Therefore, sourcing ingredients from LFS could be regarded as benefiting social, economic, and environmental sustainability in the foodservice industry.

Considering the sector of fine-dining restaurants, specific practices of sourcing ingredients from LFS have been identified. For example, chefs source local food from local suppliers or producers so that the products can be evaluated before procurement, in terms of quality, timely delivery, the ability to support required volume, consistency of the products, and price [55], as well as ability to enhance the consumption values of customers and destination branding [56]. Further, the relationship between suppliers and chefs could be more than sellers and buyers, as some restaurants have also established direct market connections with farmers and influence agricultural practices, which is conceptualised as the farm-to-table model [57]. Farm-to-table activities could further trigger agritourism and revitalise the local community by creating local employment, preserving the ethno-culinary heritage, and promoting food localism and sustainable agriculture [58]. In addition, some restaurants also have practised food foraging which is the finding and harvesting of perceived "wild" foods [59]. Food foraging ties the restaurant to the seasons and the land and reflects the historical and cultural foundations of place [21]. Even though foraging might cause some potential harm to some target species and other undesirable damage to the natural environment, depending on the species and ways of foraging, it is still often regarded as a sustainable practice in food procurement if done appropriately, because people can understand culinary heritage not only with what is on the plate but how it was harvested [59,60]. Some restaurants offer foraging courses to their customers and promote foraging tourism, which is positioned within the contexts of sustainable tourism, local food, and slow travel, and is perceived to enable more meaningful connections with place [61].

*2.3. Sustainable Preparation*

The sustainability of food preparation involves three principal considerations, including how the meal components are chosen and put together, the use of carbon-intensive foodstuffs, and how waste is managed [62]. Numerous studies have identified sustainable practices in food preparation, like purchasing renewable energy, using energy-efficient cooking routines, putting dishes on the menu that use less meat and more vegetables, preparing meals only after orders have been placed, planning purchases to avoid waste, nose-to-tail/root-to-leaf eating, reducing disposables, fermenting excess foodstuff, and composting food waste [2,6,7,62]. These practices usually contribute to environmental sustainability.

Sustainable preparation practices could be generalised in three aspects, including efficient restaurants, sustainable menus, and minimising waste. Efficient restaurants are a concern because restaurants are among the most energy-intensive types of commercial buildings, and the largest portion of energy cost is consumed by cooking and food preparation [2]. Having efficient buildings, HVAC (lighting, heating, ventilation, and cooling), kitchen appliances, refrigeration, cooking equipment, and water supply can contribute to energy conservation and pollution prevention [63,64]. Staff education and training on practising sustainable preparation and implanting habits of energy saving can also help achieve greater efficiency [65].

Sustainable menus are also significant, as the menu is a potential key factor in informing and influencing consumption decisions [43]. Providing a separate sustainable menu could thus encourage more sustainable food choices and reduce the consumption of energy-intensive foods [66,67]. A sustainable menu could be identified with such elements as vegetarian/vegan alternatives [68], carbon footprint labels [43], local food labels [69], and rankings of general environmental impacts [41]. In terms of minimising waste, food waste and disposables (e.g., glass, paper/cardboard, plastics) produced by restaurants have caused environmental issues, particularly with respect to emissions and land fill [70]. Food wastage can occur during procurement and storage (suboptimal food), production (unsold meals and meal parts), and consumption (plate waste) [71]. In fine-dining restaurants, food waste is more likely to be generated from food preparation than from customer plates due to quality assurance and aesthetic reasons [55]. Thus, restaurants and chefs have implemented nose-to-tail/root-to-leaf eating [72], composted inedible food waste [73], donated to food banks [74], and fermented extra ingredients for food (e.g., cheese) [75] or feed (e.g., fish feed) [76]. With respect to disposables, restaurants could implement selective collection, minimise packaging or use returnable packaging boxes, reduce plastics, reuse and recycle glass, and use recycled or FSC-certified paper [7].

*2.4. Sustainable Presentation*

Michelin-starred restaurants are usually regarded as luxury restaurants [34]. Previous research has examined customers' perception of sustainability in luxury restaurants [37,48,77] and Michelin-starred chefs' motivations in promoting a sustainable food experience [46]. According to these studies, sustainability could be regarded as potentially growing in importance in luxury dining consumption [47], although research remains relatively limited. As Michelin-starred restaurants could influence not only their existing or potential customers but also the actors in the food system and the public because of their high-profile reputation [32,78], their sustainable initiatives can have more profound impacts beyond the restaurant industry. Therefore, this study investigates the presentation of sustainability targeting a broader audience (i.e., suppliers, restaurants, customers, public) through three practices, including the restaurant's history, community outreach, and cooking school.

For restaurant history, some Michelin-starred restaurants display their historical heritage to boost the attractiveness of the destination that could benefit social and economic sustainability [32]. A restaurant's history could reflect the location's climate, geography, culture, history, and traditions, which further contribute to establishing the gastronomic identity of a region or locale [2,4]. Within the perspective of social sustainability, storytelling of the historical contents of the restaurants could ensure a continuation and recognition of a locale's gastronomic identity and preserve "the historical integrity of the what, how, when, and why of eating and drinking" [79]. Community outreach could reinforce social sustainability [7]. Many restaurants see themselves as part of a local community rooted in their regions, relying on local products, and representing and shaping the place [24]. Restaurants' community outreach approaches can encourage sustainable community-based activism and usually aim to reduce the stigma of specific groups, mitigate food waste with campaigns, help the disadvantaged, and highlight political and social issues [2]. Scholars have proposed some community outreach practices in the restaurant sector, like designing strategies to support their communities, donating to food banks or charities, promoting healthy eating education for the local community, and collaborating with charitable foundations or social enterprises that provide social impact [7]. For cooking school, it could be regarded as a site of living history in its staging and touristic experience of authenticity, and customers could not only eat food but also learn to identify and shop for raw ingredients, to master techniques of preparation and cooking, to become involved in local everyday life, and to understand many of the cultural beliefs behind foodways [80,81]. Cooking schools initiated by restaurants could also combine educational campaigns with dedicated training [82]. A growing inclination to incorporate sustainable culinary practices has been

introduced in culinary education and delivered by cooking schools [83], which could enhance environmental sustainability. Cooking schools may also be operated as a project in restaurants to enhance economic sustainability of both the restaurant and the surrounding region [80].

## 3. Method

Restaurants usually operate their official websites as a marketing tool for providing rich information to customers [84]. This study thus conducts a website content analysis of M3SRs' official websites to examine their sustainable procurement/preparation/presentation practices. This method is appropriate for compiling a complete list of attributes for evaluation and for studying the online content of destination restaurants, as this sector has a substantial online presence [25,85,86].

This study employed the list of Michelin three-star restaurants (M3SRs) as of 30 November 2022 as the sample. The Michelin Guide is considered one of the most authoritative indicators in the global gastronomy industry, wielding both symbolic and material power and accepted by most high-level chefs in the field [36,87]. According to the rating system of the guide, three stars mean "exceptional cuisine that is worth a special journey" [49], and M3SRs, the top fine-dining restaurants throughout the world, representing the cutting edge of culinary arts, could be regarded as the destination rather than being a stop on the way to a destination [88]. At the time of analysis (30 November 2022), 141 restaurants have been awarded three Michelin stars. 135 official websites are analysed, with six having no website. The geographical spread of M3SRs is illustrated in Table 1, and the restaurants sample is attached in Supplementary Material.

**Table 1.** The geographical spread of M3SRs with websites (*n* = 135).

| Location | *n* | % |
|---|---|---|
| France | 31 | 22.96% |
| Japan | 17 | 12.59% |
| Spain | 13 | 9.63% |
| USA | 13 | 9.63% |
| Italy | 12 | 8.89% |
| Germany | 9 | 6.67% |
| United Kingdom | 8 | 5.93% |
| Hong Kong | 7 | 5.19% |
| Switzerland | 4 | 2.96% |
| Belgium | 3 | 2.22% |
| Singapore | 3 | 2.22% |
| Macau | 3 | 2.22% |
| China Mainland | 2 | 1.48% |
| Korea | 2 | 1.48% |
| Denmark | 2 | 1.48% |
| Netherlands | 2 | 1.48% |
| Sweden | 1 | 0.74% |
| Norway | 1 | 0.74% |
| Taipei | 1 | 0.74% |
| Austria | 1 | 0.74% |

Nine sustainable practices (i.e., procurement: local food, farm-to-table activity, food foraging; preparation: efficient restaurant, sustainable menu, minimising waste; presentation: restaurant's history, community outreach, cooking school) are identified and derived from the previous literature. The website contents of each M3SRs are analysed to determine what sustainable restaurants put forward to attract customers and how the identified sustainable practices are communicated online. The criteria for determining whether a restaurant is representative of each element is defined and coded as noted in Table 2.

**Table 2.** Criteria definition and coding (*n* = 135).

| Categories | Sustainable Practices | Definition | Coding |
|---|---|---|---|
| Procurement | Local food | Does the website mention that the restaurant sources local/regional foodstuffs or equivalent from food producers? | Yes/No |
| | Farm-to-table activity | Does the website mention that the restaurant affects food production? | Yes/No |
| | Food foraging | Does the website mention that the restaurant uses foraged or wild food? | Yes/No |
| Preparation | Efficient restaurant | Does the website mention that the restaurant has an efficient building for energy/water/electricity conservation or equivalent? | Yes/No |
| | Sustainable menu | Does the website display a separate sustainable menu (e.g., vegetarian, vegan, climate-friendly, low-carbon, low footprint menu)? | Yes/No |
| | Minimising waste | Does the website mention the practice(s) to minimise waste? | Yes/No |
| Presentation | Restaurant history | Does the website mention the restaurant's history as a cultural heritage? | Yes/No |
| | Community outreach | Does the website mention the practice(s) to support the local community? | Yes/No |
| | Cooking school | Does the website mention that the restaurant has a cooking school to deliver culinary philosophy and cooking methods? | Yes/No |

## 4. Results and Discussion

The percentages of the practices mentioned on M3SRs' official websites are indicated in Table 3. From the results, identified sustainable practices are mentioned by less than 50% of M3SRs with the highest percentage being local food (42.96%) and the least percentage being food foraging (5.19%).

**Table 3.** The sustainable practices of Michelin three-star restaurants (*n* = 135).

| Categories | Sustainable Practices | *n* | % |
|---|---|---|---|
| Procurement | Local food | 58 | 42.96% |
| | Farm-to-table activity | 18 | 13.33% |
| | Food foraging | 7 | 5.19% |
| Preparation | Efficient restaurant | 12 | 8.89% |
| | Sustainable menu | 23 | 17.04% |
| | Minimising waste | 18 | 13.33% |
| Presentation | Restaurant history | 47 | 34.81% |
| | Community outreach | 23 | 17.04% |
| | Cooking school | 16 | 11.85% |

For research question one, "the extent to which sustainable practices are promoted as part of MSRs' offerings", the results indicate that sustainability has not been considered a necessary offering in M3SRs or a keyword in their website marketing and further reflect M3SRs' failure to embrace sustainability, which is inconsistent with their roles as sustainability ambassadors that previous studies suggest [37,46,89]. This phenomenon has some possible explanations. First, some websites are used as online reservation platforms with limited information to introduce the restaurant philosophy or practices, including sustainability, such as Zén in Singapore, Sushi Yoshitake in Japan, and Benu in the USA. Second, providing a luxury dining experience appears to be incompatible with sustainability, given that many fine-dining restaurants keep searching for luxury and off-season ingredients from distant regions [48] and produce much more food waste than other types of restaurants just due to aesthetics concerns [90]. Third, this study argues that the current model for fine-dining restaurants seems to go the opposite way, far from sustainability. Food critics have begun to question the sustainability of fine-dining restaurants and their performance of a rococo act for a rarefied audience that is "forever trying to dazzle self-regarding epicures with new stunts, novel sensations, modes of presentation we hadn't imagined, flora and fauna rarely pinned down on a plate" [91]. Some recent news has also reinforced the criticism, including Noma in Copenhagen announcing that they will close the current restaurant in 2025 and "create a lasting organization dedicated to ground-breaking work in food" [92], Amass in Copenhagen, which was a true leader in sustainability with a Michelin

Green Star, being taken under bankruptcy [93], as well as the release of a satirical film, *The Menu*, which interrogates the culture created by fine-dining restaurants and chefs [94].

Nevertheless, the rest of M3SRs who provide sustainability-related information on their websites are still worth investigating because they seem to demonstrate sustainability online in a way that is acceptable by the Michelin Guide judges, food critics, foodies, and the restaurant industry [46] and act as pioneers in leading food trends in the food system by creating food media and cookbooks [32]. Therefore, to address research question two, "how the identified sustainable practices reflect the locality", this study identified nine sustainable practices in M3SRs that promote locality to bridge sustainability and luxury dining experience, potentially providing insights into the compatibility between the luxury dining experience and sustainability, as well as the possible influence of M3SRs' as culinary leaders on the foodservice industry and other types of restaurants in practising sustainability.

### 4.1. Local Food

Most identified M3SRs communicate their food procurement strategies of locally-sourced food on their website (42.96%). Introducing local foodstuffs implies high food quality, as local food is usually perceived as fresh and in season [24] and could be regarded as a sustainable practice benefiting the locality. For environmental sustainability, L'OSIER in Japan sources food from local farms and mentions that "Chef Olivier Chaignon visited Kozagawa . . . Here, they harness symbiosis between diverse microorganisms to develop their own soil, growing vegetables and edible roses using natural farming techniques, without the use of pesticides, chemical fertilizers, or animal fertilizers . . . young staff members were making a greenhouse out of bamboo and reused construction materials" [95], which indicate the conservation of soil. Azurmendi in Spain works on a germplasm bank, which is "a program on hydroponic crops with local varieties of vegetables in danger of extinction. Currently, they are working on . . . It hosts more than 400 local seed varieties of vegetables and aims to show the importance of preserving genetic diversity" [96], to practise the conservation of local biodiversity. For social sustainability, Alléno Paris au Pavillon Ledoyen in France began a series of experiments to ferment local food and promote "terroirs as seen through the lens of fermentation . . . namely the 'gastronomisation' of the terroir" [97]. This initiative enriches local culinary culture by privileging the centrality of the relationship between food, culture, history, and geography and could further boost culinary tourism [21,44]. For economic sustainability, non-agricultural activities, such as culinary tourism, that are associated with the local food system can benefit local economic growth and local employment [20,98]. Épicure in France "fosters relationships with local and artisanal suppliers to support the health of our local economies" [99]. Akelaŕe in Spain "works as much as possible with local and national suppliers, reducing the transportation of material and waste and supporting the creation of employment at the local level" [100]. Les Prés d'Eugénie—Michel Guérard in France indicates that "we work directly with producers, farmers and butchers, not with wholesalers. This ethical choice guarantees the highest quality and fair remuneration for the farmers" [101]. Despite the percentage of local food being the highest in all identified sustainable practices, less than half of M3SRs indicate this initiative on their websites, which suggests that promotion of local produce has not been fully incorporated with mainstream fine-dining restaurants. Compared to the characteristics such as rare foodstuffs or complicated cooking techniques, local produce may not seem to display the exclusiveness and the premium price of luxury fine dining [102].

### 4.2. Farm-to-Table Activity

13.33% of reviewed M3SRs indicate their farm-to-table activity on their official websites. This practice requires restaurants to establish restaurant-farmer relationships and foster sustainable agriculture systems [57], which is difficult and could be regarded as a reason for the low percentage of farm-to-table activity. Nevertheless, this study finds that some M3SRs indicate their farm-to-table activity in food procurement and highlight

a win-win partnership that could benefit not only the local food system but also the local community. For example, Quince in the USA cooperates with a local farm and indicates that "the seasonally-changing menu at Quince stems from this close collaboration and partnership between Peter, who operates Fresh Run Farm, Chef Michael Tusk and the Quince culinary team ... Fresh Run Farm grows conscientiously and organically to safeguard this agricultural resource for future generations to come" [103]. This example thus illustrates the role of farmers in participating in the restaurant's menu design and spotlighting local culinary culture. Single Thread in the USA has a team to "work the fields by hand to minimise our impact on the land ... We grow hundreds of different varieties of culinary crops throughout the seasons as well as cut flowers with care and integrity ... We remain optimistic for a world with a stronger emphasis on seasonality and locality and humbled by the ever changing climate" [104]. Thereby, it highlights the restaurant's contributions to local agriculture development and ecosystem conservation. Similarly, Le Clos des Sens has "create a natural permaculture ... , which is the contraction of the English words 'permanent' and 'agriculture' ... , and is a complete spiritual path (Tao) based on 'inaction' with four principles of no ploughing, no fertiliser, no weeding and no pesticides" [105]. Interestingly, Chez Panisse in the USA is famous for creating the farm-to-table model, spurring the slow, local, and organic food movements since 1971 [57], although its website has little information related to the farm-to-table movement.

*4.3. Food Foraging*

Only seven M3SRs mentioned that they forage wild food, and the percentage (5.19%) is the lowest in the reviewed sustainable practices. Highlighting foraged food is a significant characteristic of terroir restaurants used to demonstrate their authenticity [59,106]. This study finds that food foraging can benefit locality in terms of local seasonality, culinary culture, and biodiversity. Noma in Denmark, with Chef René Redzepi, is a strong promoter of regional gastronomy [31] and indicates that "our origin is rooted in an exploration of the natural world, which began with a simple desire to rediscover wild local ingredients by foraging and to follow the seasons" [107], which highlights the respect for local seasonality. St. Hubertus in Italy launched a food foraging project, namely "St. Hubertus Unplugged at 2000 m". "Guests and chefs will begin the journey and hike up towards our cabin, foraging wild herbs, mushrooms and other ingredients that will then be used to prepare an unforgettable lunch on the wooden fire" [108]. The guests could thus understand the restaurant's "food concept of 'Cook The Mountain' with its origins" [108], which could further enhance local culinary culture conservation and boost rural/culinary tourism. Further, foraging wild food could also contribute to local biodiversity. Aponiente in Spain found Zostera marina on their usual underwater expeditions, "which is one of the four marine phanerogam species that exist in Cadiz and that currently grow naturally". And then "the Aponiente Gastronomic Research Laboratory has managed to grow Zostera marina and obtain its most coveted product, the marine cereal, in a controlled way for the first time in history" [109] (p. 12). Considering the low percentage of foraged food, several reasons can be put forward as to why M3SRs may find difficulties in incorporating such practices: (1) It is difficult for the M3SRs located in developed cities to access wild food as foraging may be prohibited on public lands [110]; (2) Food foraging may have limited consumer appear as it is usually employed to highlight food locality in terroir restaurants or create authentic guest experiences [21,111]. Given that most of M3SRs do not position themselves as terroir restaurants, food foraging may not be regarded as a necessary offering or item to promote.

*4.4. Efficient Restaurant*

Only thirteen M3SRs (8.89%) introduced the efficiency of the restaurants on their websites in terms of buildings, energy consumption, and staff education. Some restaurants demonstrate their efficient building that is environmentally friendly. For example, Azurmendi in Spain is in a bioclimatic building, "incorporating non-invasive methods of

working with the environment, local and recycled materials and cutting-edge technology in terms of renewable energy" [96] (p. 5). Akelaŕe in Spain mentions that "our commitment to the environment begins with the integration of the building into its surroundings, using natural raw materials and covering plants that reduce the visual impact, in addition to serving as a natural insulator" [100]. Similarly, Schwarzwaldstube in Germany makes the roof as "the home and living environment for different insects, which are very important for our ecosystem" and "a cleaner for the rainwater and with it the wastewater is balanced" [112]. These sustainable initiatives could thereafter enhance the local conservation of energy, water, and electricity. With regards to the staff, Atelier Crenn in the USA indicated that "our staff are continually trained on our strict standards of recycling and our plastic-free future" [113], which could ensure the efficiency of staff practising sustainability. All of these efficient restaurant initiatives could contribute to local environmental sustainability. However, two reasons can be provided as to why few M3SRs indicate their sustainable practices for an efficient restaurant on official websites: (1) Such energy-saving kitchen appliances and staff education might not have significant influences on the restaurant's service quality or attractiveness to customers; (2) These initiatives might not be perceived and evaluated by customers directly [2].

### 4.5. Sustainable Menu

Many M3SRs do not display their menu on their websites. It is possible that not disclosing menu details can also be regarded as an alternative marketing strategy to spark further interest in consumers and suggest freshness and daily determination of what is served, especially in M3SRs which usually require reservations more than six months in advance [25,114]. In total, 17.04% of reviewed M3SRs design a separate sustainable menu for their customers and present it on their official website. Most of them regarded a vegetarian set menu as a sustainable alternative. For instance, The Inn at Little Washington in the USA names their vegetarian creations "The Good Earth Menu" [115]. Hof van Cleve in Belgium describes their vegetarian menu as "Field, garden and wood" [116]. L'Oustau de Baumanière in France mentions that "we have created a menu for you featuring fresh vegetables from our gardens and neighbouring farms. To present them cooked in different ways, we have chosen to accompany them with the finest virgin olive oils from the Baux Valley Appellation" [117]. Within the description, not only are the locally sourced vegetables highlighted, but also the local speciality of olive oil.

### 4.6. Minimising Waste

About 13.33% of reviewed restaurants mention their sustainable practices on minimising waste that could benefit their environmental sustainability, including reducing, reusing, recycling food waste and disposables; and some of them also indicate such practices' contributions to the locality. For food waste, as noted above, St. Hubertus in Italy promotes the culinary philosophy of "Cook the Mountain", which has "a crucial point is the approach of no waste, to try to avoid it buying only the necessary and changing the consumer behaviour. In that way, we can avoid leftovers, the excess of food and we treat the natural resources in a responsible way" [108]. Schwarzwaldstube in Germany "use many regional ingredients, in line with the philosophy 'from nose to tail, from root to flower' . . . . In this way, we are also expressing our responsibility for the beauty of the Black Forest in the kitchen and making a small contribution to enabling the next generation to enjoy it too" [112]. Such initiatives highlight reducing food waste generated before consumption, which may be more significant in fine-dining restaurants than casual restaurants because of the need to assure quality and aesthetics [90]. Some other M3SRs also mention dealing with food waste after consumption, like composting to fertilise their local gardens/farms, such as Atelier Crenn in the USA, Da Vittorio in Italy, and Arpège in France. As for disposables, Christophe Bacquié in France indicates that "the caps from plastic water bottles are given to the association 'Bouchons d'amour' to be recycled into medical equipment [118]. Cenador de Amós in Spain "chooses an eco-friendly rug created from plastic and the remains of

fishing nets collected from the ocean" [119] (p. 25). Few M3SRs provide information about disposables, and a possible explanation might be that fine-dining restaurants usually only provide on-premises dining to ensure the customers' dining experience, so that fewer single-use packages and tableware items for food delivery or takeaway are used in fine-dining restaurants compared to others [120]. However, offering takeaway and home delivery became a new normal in fine-dining restaurants post-pandemic [17,121]. Therefore, the sustainability and locality of food delivery and takeaway should be further considered in the fine-dining industry, especially in terms of disposables.

### 4.7. Restaurant History

Given that M3SRs could be regarded as a destination within a place [49], 34.81% of them introduce their history on websites to promote their culture and attract customers. This study finds that restaurant history can represent local culture in three ways, including local culinary traditions, local architecture, and the local community, which are small and subtle enhancements to existing local food culture [25]. Additionally, the story of a restaurant could forge a relationship between the destination and customers who might become strong brand advocates and loyal buyers, benefiting social and economic sustainability [122]. For local culinary traditions, Kashiwaya in Japan "specialises in kaiseki cuisine which remains faithful to Japanese culinary tradition" [123]; Gaon in Korea lists the history, with subtitles including "the first to take not of 'the possibility of Korean Cuisine', 'the dream of the globalisation of Korean cuisine' cautiously blossoming, Korean cuisine enrapturing epicures around the world, the essence of Korean culture, truly 'the centre of Korean cuisine"; Pic in France was first founded in 1889, "since then, through 3 generations and their 3 stars, the history of the PIC house is intimately tied to French gastronomy and embodies the French art of living" [124]. For local architecture, The French Laundry in the USA details the history of the building from 1900 to today [125]; Villa Crespi in Italy introduces the villa and refers to the fact that "dating back to the late 19th Century, this Moorish style, historic residence has been a private home and over the centuries became an exclusive place of hospitality for poets, nobles and rulers. Its name resonated in some of the most illustrious houses in Northern Europe" [126]. For local community, St. Hubertus in Italy highlights the community of Alta Badia, which "is one of five Ladin-speaking valleys in northern Italy", and suggests that "the shared passion for our territory is where the mutual exchange with our farmers begins: the mission is to cultivate the diversity of our cultural landscape in order that everyone can benefit" [108]. Moreover, the local historical culture could be represented by other forms, like the decorations in the restaurants, such as René et Maxime Meilleur in France decorating their interior environment with "the China cabinet which proudly displays a collection of ancient regional dishes", "the ancient dishes and pottery—some from the 18th century—the covers marked with the historic coats-of-arms of Savoie", "curtains, sofas, chairs and uniforms which use the precious Bonneval wool and fabric made by the 200-year-old Arpin spinning mill with ancestral know-how" [113].

### 4.8. Community Outreach

In total, 17.04% of identified M3SRs demonstrate sustainable practices that benefit the community. This study finds that many of them are targeted to the local community for supporting vulnerable groups, local businesses, and local transformation. For vulnerable groups, Eleven Madison Park in the USA indicates that "every dinner purchased at Eleven Madison Park allows us to provide meals to these New Yorkers experiencing food insecurity . . . to combat food insecurity and ultimately, create a more sustainable and equitable food system while utilizing our platform to spread awareness of the issue" [127], which reinforce local social sustainability. Similarly, Sketch (The Lecture Room & Library) in the UK is involved in philanthropy with Place2Be to "deliver emotional and therapeutic services in primary and secondary schools, building children's resilience through talking, creative work and play" [128]. For local producers, Le Petit Nice in France "works with the

fishermen, ever since Germain Passedat founded the establishment, and Gérald Passedat has refined his collaboration with local fishermen by drawing up a set of specifications respecting sizes and seasons, and laying claim to traditional fishing techniques" [129]. With more fisherpersons practising sustainable fishing, the restaurant's initiative could benefit local environmental sustainability. In terms of local transformation, Aponiente in Spain "is not just a restaurant, but an ambitious project whose aim is to reactivate and recover the environment surrounding the restaurant, by restoring the ecosystem and re-establishing the natural balance, taking care of the natural capital of the marshes to promote the ecosystem services, obtain foods due to its rich biodiversity, and as a source of income, wealth and employment for the area" [109] (p. 7). Such projects could further contribute to local environmental and economic sustainability.

*4.9. Cooking School*

There are sixteen (11.85%) M3SRs that mention their cooking schools. Apart from normal master courses offered for professional chefs, some M3SRs also provide opportunities for their customers to participate in food procurement and preparation processes. For example, De Librije in Netherlands offers a vegetarian cuisine workshop and "works with vital vegetables from our local supplier Eef Stel" [130], which could not only promote a more sustainable dining choice of vegetarian diets but also enhance customers' knowledge of local vegetarian products, benefiting environmental sustainability in the long term. Da Vittorio in Italy guides the customers to explore the location; "food enthusiasts can attend beginners and advanced courses, single-theme events, excursions to markets and production sites, tastings and discussions" [131]. Akelaŕe in Spain "hosts Master Classes in which we introduce our guests to the ingredients and techniques involved in the preparation of the dishes they will afterwards enjoy in the dining room" [100]. Additionally, some restaurants also share their creations in cooking school, which could further enrich the local culinary culture and enhance local social sustainability. For instance, Alain Ducasse in France mentions that "as co-founder of the French Culinary School, Alain Ducasse stands for the idea that the influence of French cuisine is one of France's greatest assets as a primary, worldwide tourist destination . . . to celebrate the identity, vitality, tradition and diversity of French cuisine and lifestyle" [132]. This study finds that cooking schools initiated by M3SRs usually target two types of audiences: (1) Restaurant customers or tourists could attend such classes to spend time in the domestic environment and with local people, experiencing authenticity and locality [80]; (2) Professional chefs and chef apprentices could also be educated in cooking schools organised by these word-leading restaurants [133]. However, it is notable that few studies have researched cooking schools as a part of restaurants' offerings. This study thus calls for more research on cooking schools initiated by these high-profile restaurants.

**5. Conclusions**

Sustainability has been widely discussed in relation to the contemporary food system, and various sustainable practices have been promoted in the restaurant and foodservice industry [2]. However, some sustainable practices may be regarded as difficult to implement, especially following the impacts of COVID-19. For example, Le Petit Nice in France claims that "between economy (ensuring fishermen's daily livelihood) and ecology (conserving resources), sustainable traditional fishing is a difficult balance, and the relationship between gastronomy and environmental responsibility is sometimes strained: it can be complicated" [129]. Existing literature on sustainable restaurants also puts more emphases on ecological sustainability rather than social and economic sustainability [4]. Facing such a dilemma, this study searches for more possible ways to sustain sustainability in the restaurant industry. Therefore, this study reviews and examines sustainable practices during the processes of procurement, preparation, and presentation at M3SRs. By conducting a content analysis of the official websites of 135 M3SRs, this study finds that most of the sustainable practices could be embedded with the locality and benefit local ecological,

social, and economic sustainability. From the results, procuring foodstuffs from local food systems and introducing the restaurant's history as a part of local cultural heritage could be more common practices for restaurants to use to enhance sustainability, while foraging local food and improving the restaurant's efficiency are less mentioned in the identified M3SRs' websites.

Embedding the sustainable practice in the locality could sustain sustainability from environmental, social, and economic perspectives in the restaurant industry. Procuring local food, incorporating farm-to-table activity, foraging local food to protect biodiversity, enhancing restaurant efficiency, designing sustainable menus, minimising waste, and promoting sustainable diets in cooking schools could contribute to local ecological sustainability. Similarly, collaborating with local food suppliers, training staff to help improve restaurant efficiency, introducing a restaurant's history as local culture, local community outreach, and presenting local culinary culture in cooking schools could reinforce social sustainability. Finally, working closely with local food purveyors, developing local tourism by promoting local culinary culture, and financially supporting local community could thereafter enhance local economic sustainability. Furthermore, it is also suggested that before investing in the advanced energy-saving equipment and pursuing green stickers, sustainable restaurants could be more engaged in the local community, promoting locality, and thereby contributing to the sustainability of the place in which they are situated.

This study also highlights the debates between globality and locality that still exist in the foodservice industry, especially in higher-end restaurants that focus on particular cuisine styles. For example, Memories in Switzerland indicates that "It's important to me to combine our local focus with a cosmopolitan outlook" [134]. However, most of the M3SRs mentioned in the discussion appear determined to become localised terroir restaurants. St. Hubertus in Italy also indicates the transformation from globality to locality, "a radical and courageous decision was made after careful consideration in 2011: Classic gourmet cuisine with fine products from all over the world was replaced—the new leitmotiv was territoriality and seasonality in the consciousness of our old traditions from that moment onward" [108]. In many cases, these M3SRs would be regarded as destination restaurants that attract both locals and culinary tourists [25,135]. Importantly, the locality that is regarded as a part of the restaurant's offerings can not only benefit the environment but also provide customers with a sense of place, and the value of originality and authenticity [136]. They can also be regarded as a tool for local development, involving local restaurants, farmers, farmer's market vendors, and wholesale distributors [24].

In terms of theoretical implications, fine-dining restaurants' sustainable practices are reviewed and identified in terms of environmental, social, and economical sustainability. This study also bridges sustainability and locality in the fine-dining restaurant sector and emphasises the importance of locality in the promoted sustainability practices. With respect to practical implications, this study focuses on the sustainable practices in fine-dining restaurants and suggests that these restaurants and other foodservice providers regard locality as a direction for restaurant sustainability and connectedness to its surroundings in the long term. A further point of study here is whether such connectedness may also enhance restaurant business and community resilience, as has been suggested elsewhere [2].

This study has limitations in that it only analysed the sustainability of M3SRs by official website content, but not researching other types of fine-dining restaurants and other information on sustainable practices. Some restaurants do not renew their website to date, and some even do not have official website. Therefore, some practices are missed due to the limited information on their official websites. On the other hand, whether the sustainable practices that are discussed on their websites are conducted in the long term or not remains unknown. Future research could therefore include interviews, surveys and/or observation, and fieldwork to be able compare what is stated on the websites against actual practices, as well as to identify the reasons for the online marketing strategy and content. The awarded M3SRs are all fine-dining restaurants evaluated by the criteria of the Michelin Restaurant Guide, and most of them are in European countries due to the Michelin Guide's focus



and criteria. Therefore, the linkages between sustainability and locality examined in this study only reflect the situations of these fine-dining restaurants. Future research would be fruitful in examining other types of restaurants (e.g., casual restaurants, chain restaurants) or restaurants awarded other sustainability certifications, as compared to the Michelin three-star restaurants, especially so as to enlarge the geographical and gastronomic scope of how a sustainable restaurant contributes to the locality.

**Supplementary Materials:** The following supporting information can be downloaded at: https://www.mdpi.com/article/10.3390/su15043672/s1, Table S1: Details of the Michelin Three-Star Restaurants.

**Author Contributions:** Conceptualization, methodology, Y.H. and C.M.H.; formal analysis, writing—original draft preparation and editing, Y.H.; writing—review and editing, supervision, C.M.H. All authors have read and agreed to the published version of the manuscript.

**Funding:** This research received no external funding.

**Institutional Review Board Statement:** Not applicable.

**Informed Consent Statement:** Not applicable.

**Data Availability Statement:** The data presented in this study are available on request from the corresponding author.

**Conflicts of Interest:** The authors declare no conflict of interest.

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
