# Peer review of "Locality in the Promoted Sustainability Practices of Michelin-Starred Restaurants"

_sustainability, doi:10.3390/su15043672_

Round 1
Reviewer 1 Report
This is an interesting paper with a lot of potential. I read the article with high interest. However, it needs some more work before it can be ready for acceptance. Although the quality of writing is generally good, the paper would benefit for much more carefully address the issues mentioned at below.
Abstract:
· The abstract is beautifully written, and the purpose of this study is clear. However, methodology and the limitation of this study needs to be incorporated in the abstract section.
Introduction:
· The introduction section is through and well organized.
Review of the Literature:
· The literature review section is through and well organized.
Method:
· The method section is very weak. The author(s) used the content analysis however, it is necessary for how the content analysis has been conducted or the procedure needs to be mentioned. For example, how the textual data has been evaluated. Why the content analysis? Main advantage of content analysis, etc. May be a separate section with “Content Analysis” heading would be beneficial.
Results and Discussions:
· The results section is extremely limited. The author(s) need to do a better job for describing the whole results. Many results area is incorporated in the discussion section and that can be shifted to the results section. Or can be done both results and discussion section at the same place and same time.
· The discussions should be tied up with more literature from the literature review sections.
Conclusions:
· There are no practical and theoretical contribution in this paper. Author(s) should have separate theoretical and practical implications section.
Suggestion:
· Overall, the language of this paper is noticeably clear, and the paper is written very well. However, there are discrepancies among the literature review, results, and discussion sections. The entire manuscript needs to be tied up in logical way.
· Addressing the above issues will enhance your work. I am looking forward to reading your revised paper. Good luck to your work.
Author Response
We would like to express our gratitude to the editors and the reviewers for their helpful comments and suggestions for improving the quality of our paper. We have revised and rewritten our paper accordingly. All major changes have been highlighted in red in the document.
Point 1: Abstract: The abstract is beautifully written, and the purpose of this study is clear. However, methodology and the limitation of this study needs to be incorporated in the abstract section.
Response 1: We appreciate the reviewer’s positive comment. We have added the methodology “This study examines the sustainability of 135 Michelin three-star restaurants by conducting website content analysis”, and limitation “Although this study is limited to the website content of Michelin three-star restaurants by official websites it provides potentially valuable insights on the promotion of sustainable restaurant practices” in the Abstract.
Point 2: Introduction: The introduction section is through and well organized.
Response 2: We appreciate the reviewer’s positive comment.
Point 3: Review of the Literature: The literature review section is through and well organized.
Response 3: We appreciate the reviewer’s positive comment.
Point 4: Method: The method section is very weak. The author(s) used the content analysis however, it is necessary for how the content analysis has been conducted or the procedure needs to be mentioned. For example, how the textual data has been evaluated. Why the content analysis? Main advantage of content analysis, etc. May be a separate section with “Content Analysis” heading would be beneficial.
Response 4: According to the reviewer’s suggestions, we have revised Method section by explaining why website content analysis is applied in the first paragraph of Method, “Restaurants usually operate their official websites as a marketing tool for providing rich information to customers. This study thus conducts a website content analysis of M3SRs’ official websites to examine their sustainable procurement /preparation /presentation practices. This method is appropriate for compiling a complete list of attributes for evaluation and for studying the online content of destination restaurants, as this sector has a substantial online presence “(Line 263-268). We have also detailed the criteria definition and coding of identified sustainable practices at the end of Method section, “The website contents of each M3SRs are analysed to determine how sustainability restaurants put forward to attract customers and how the identified sustainable practices are communicated online. The criteria for determining whether a restaurant is representative of each element is defined and coded as noted in Table 2.” (Line 285-291).
Point 5: Results and Discussions: The results section is extremely limited. The author(s) need to do a better job for describing the whole results. Many results area is incorporated in the discussion section and that can be shifted to the results section. Or can be done both results and discussion section at the same place and same time.
The discussions should be tied up with more literature from the literature review sections.
Response 5: As the reviewer suggests, we have merged Results and Discussion section. We have also explained why some percentages are low and tied up findings with literature. For example, from Line 294 to 301, we have explained why all sustainable initiatives are mentioned by less than half M3SRs, “Some possible explanations would be that 1) promoting sustainability might be incompatible with providing fine-dining experience because importing off-season foodstuffs could not be sustainable; 2) some websites are just used as online reservation platforms with limited information to introduce their sustainable initiatives” From line 335-341, we have mentioned that local food has not been incorporated with mainstream fine-dining restaurants because it seems not enough to display the exclusiveness and the premium price of fine dining. Similar explanations in other sustainable practices are also marked in red, including farm-to-table activity (Line 344-346), food foraging (Line 387-394), efficient restaurant (Line 412-416), sustainable menu (Line 418-421). We have also clarified our findings that could support previous studies on minimising waste, like food waste are generated before consumption in fine-dining restaurants (Line 444-446) and disposables in fine-dining restaurants are less studied while the use of disposables has been increased (Line 452-459). More detailed discussions are also provided for restaurant history (Line 463-468), and cooking school (Line 533-540).
Point 6: Conclusions: There are no practical and theoretical contribution in this paper. Author(s) should have separate theoretical and practical implications section.
Response 6: Based on the reviewer’s suggestions, we have added the theoretical and practical implications part in the conclusion (Line 590-598).
Point 7: Suggestion: Overall, the language of this paper is noticeably clear, and the paper is written very well. However, there are discrepancies among the literature review, results, and discussion sections. The entire manuscript needs to be tied up in logical way. Addressing the above issues will enhance your work. I am looking forward to reading your revised paper. Good luck to your work.
Response 7: We appreciate the reviewer’s helpful comments. We have revised the article accordingly and marked the revision in red.
Reviewer 2 Report
Sustainability
Locality in the Promoted Sustainability Practices of Michelin-Starred Restaurants
Yuying Huang 1, and C. Michael Hall
Introduction
Line 38. I do not think there is a broad agreement that sustainability includes fairness among nations.
Line 60. “more appropriate ways” – are you abandoning expensive solutions?
Line 70. Locality makes sense in smaller cities and towns, but makes less sense in the growing mega-cities of 10 million plus people.
Literature review
Line 100-111. This emphasis on prestige offends some diners who want good food, not prestige.
Line 112. Are you referring to restaurants in Europe, or more globally?
Line 137. Interesting that better quality is not one of the 8 benefits of local.
Line 146. How does local food ensure product quality?
Line 204. Generation to generation – that used to be the case, now many luxury restaurants have contemporary themes and ingredients.
Line 214. This sounds like Europe.
Method
Line 239. Not sure I agree on the status of Michelin guide.
Line 266 Table 2. Procurement indeed supports sustainability. Preparation scores low but also supports sustainability. Presentation does not appear to support sustainability.
Discussion
Even when the % of M3SRs is very low for a factor the authors discuss this for the small number of restaurants doing it. They do not discuss why this % is low.
Conclusions
Line 453-458. Yes! This is the first mention of the challenges of sustainability practices. More of this needed throughout the paper.
L. 462. How do we know that official websites are valid, rather than advertising?
Overall, the paper attempts to use Michelin restaurants as a way of looking at sustainability. Unfortunately the paper depends on what is presented on restaurant websites – we have no idea whether that information is valid. The overall presentation should include both positive and negative information – for example, why do so few restaurants practice certain sustainable practices? In its current form the paper sounds like more advertising for the Michelin restaurants. And the authors need to acknowledge that many customers looking for good food are turned off by the emphasis on prestige and the practice of very high prices. Many customers prefer the authentic local restaurant run by the same people for 30 years.
This paper needs a very major revision.
Author Response
We would like to express our gratitude to the editors and the reviewers for their helpful comments and suggestions for improving the quality of our paper. We have revised and rewritten our paper accordingly. All major changes have been highlighted in red in the document.
Point 1: Line 38. I do not think there is a broad agreement that sustainability includes fairness among nations.
Response 1: According to the reviewer’s suggestion, we have revised the expression with “Previous research indicated five basic principles of sustainability… (5) the goal of achieving a better balance of fairness and opportunity (equity) between and within nations” (Line 36, 41-42) to reduce ambiguity. We agree with the reviewer’s opinion, while we suppose that it is not a key issue that related to the research problem. The five principles of sustainability are cited to emphasis the importance to research sustainability.
Point 2: Line 60. “more appropriate ways” – are you abandoning expensive solutions?
Response 2: We have revised it with “more appropriate and cost-effective ways” (Line 63) to response that some sustainable practices are too expensive for restaurants to maintain in the long term, especially when they are facing financial difficulties.
Point 3: Line 70. Locality makes sense in smaller cities and towns but makes less sense in the growing mega-cities of 10 million plus people.
Response 3: We have added a sentence before the paragraph that comparing globality and locality, “Globalisation is often regarded as a process that makes geography less relevant for social and cultural arrangements, which have impaired the diversity and quality of the local food culture. In contrast to globality, locality has become an increasingly prevalent food trend in recent years which could be regarded as a form of local cultural capital that strengthens sense of place, place and product differentiation, and place branding in a globalised world” (Line 83-88). We regard local culinary culture as a local cultural capital that would potentially be beneficial to the environmental, social, and economic sustainability. We think both small towns and mega-cities could employ the locality as cultural capital to enhance the imagery of place, even though mega-cities might have many other kinds of cultural capital, so that the locality of culinary culture for mega-cities would be less important than that for small towns.
Point 4: Line 100-111. This emphasis on prestige offends some diners who want good food, not prestige.
Response 4: We simplify the expression by “The Michelin Guide is regarded as a tastemaker of contemporary food and restaurant culture” and mentioned “high-quality food”. We also emphasis MSRs’ influence in the food system, we write: “MSRs could also lead food trends and promote culinary-related initiatives in the food system due to the high profile of Michelin Stars in food media and the wider restaurant industry” (Line 113-122).
Point 5: Line 112. Are you referring to restaurants in Europe, or more globally?
Response 5: We aim to target global restaurants in this paragraph, so we have cited more papers that studies global restaurants rather than only European restaurants (Ref. 20, 41-45).
Point 6: Line 137. Interesting that better quality is not one of the 8 benefits of local.
Response 6: We have changed the sentence, “LFS can increase consumers’ access to fresh and healthy food” (Line 159). We think “fresh and healthy” might be a more suitable way to describe the good quality of food.
Point 7: Line 146. How does local food ensure product quality?
Response 7: This sentence was not clear enough. Sourcing food locally is easier for the restaurants to evaluate food quality and select the good-quality and fresh food for local producers, so we have changed the expression, “Chefs source local food from local suppliers or producers so that the products could be evaluated before procurement, in terms of quality, timely delivery, the ability to support required volume, consistency of the products and price” (Line 167-170).
Point 8: Line 204. Generation to generation – that used to be the case, now many luxury restaurants have contemporary themes and ingredients.
Response 8: This paragraph aims to highlight the sustainability in luxury restaurants, so we have revised the sentence, “Previous research has examined customers’ perception of sustainability in luxury restaurants and Michelin-starred chefs’ motivations in promoting sustainable food experience. According to these studies, sustainability could be regarded as potentially growing in importance in luxury dining consumption, although re-search remains relatively limited.” (Line 223-228).
Point 9: Line 214. This sounds like Europe.
Response 9: We target to global restaurants that display their historical heritage as the attractiveness of the destination. In the discussion part, we have also provided some evidence in Japan, Korean, the USA. We can understand the reviewer’s concern on the preference of European restaurants in the Michelin Guide. It is true that the percentage of European restaurants is higher than restaurants in the other parts of world in the Michelin Guide due to its originality, but we cannot deny its clout in the global restaurant industry. To address the reviewer’s concern, we have mentioned this point in the limitation, “the awarded M3SRs are all fine-dining restaurants evaluated by the criteria of the Michelin Restaurant Guide, and most of them are in European countries due to the Michlin Guide’s originality” (Line 605-607).
Point 10: Line 239. Not sure I agree on the status of Michelin guide.
Response 10: The sentence was not objective enough, we have revised it, “The Michelin Guide is considered as one the most authoritative indicator in the global gastronomy industry, wielding both symbolic and material power and accepted by most high-level chefs in the field” (Line 270-272).
Point 11: Line 266 Table 2. Procurement indeed supports sustainability. Preparation scores low but also supports sustainability. Presentation does not appear to support sustainability.
Response 11: Procurement and preparation are more related to environmental sustainability, while the restaurant history is related to social and economic sustainability, community outreach is related to social sustainability, and cooking school related to environmental and economic sustainability. These sentences are added in the literature review to emphasis the presentation practices support sustainability. In the literature review section of Sustainable Presentation, we have rewritten that “For restaurant history, some Michelin restaurants display their historical heritage to boost the attractiveness of the destination that could benefit social and economic sustainability” (Line 235-237); “Community outreach could reinforce social sustainability” (Line 242-243); “A growing inclination to incorporate sustainable culinary practices has been introduced in culinary education and delivered by cooking schools, which could enhance environmental sustainability. Cooking schools may also be operated as a project in restaurants to enhance economic sustainability of both the restaurant and the surrounding region” (Line 257-261). Also, we have emphasized the sustainability enhanced by the three practices with evidence in the Discussion section, including restaurant history (Line 463-467), community outreach (Line 500, 506-508, 513-514), and cooking school (Line 522, 528-529).
Point 12: Discussion: Even when the % of M3SRs is very low for a factor the authors discuss this for the small number of restaurants doing it. They do not discuss why this % is low.
Response 12: According to the reviewer’s suggestions, we have explained why some percentages are low and tied up findings with literature. For example, from Line 294 to 301, we have explained why all sustainable initiatives are mentioned by less than half M3SRs, “Some possible explanations would be that 1) promoting sustainability might be incompatible with providing fine-dining experience because importing off-season foodstuffs could not be sustainable; 2) some websites are just used as online reservation platforms with limited information to introduce their sustainable initiatives” From line 335-341, we have mentioned that local food has not been incorporated with mainstream fine-dining restaurants because it seems not enough to display the exclusiveness and the premium price of fine dining. Similar explanations in other sustainable practices are also marked in red, including farm-to-table activity (Line 344-346), food foraging (Line 387-394), efficient restaurant (Line 412-416), sustainable menu (Line 418-421). We have also clarified our findings that could support previous studies on minimising waste, like food waste are generated before consumption in fine-dining restaurants (Line 444-446) and disposables in fine-dining restaurants are less studied while the use of disposables has been increased (Line 452-459). More detailed discussions are also provided for restaurant history (Line 463-468), and cooking school (Line 533-540).
Point 13: Line 453-458. Yes! This is the first mention of the challenges of sustainability practices. More of this needed throughout the paper.
Response 13: According to the reviewer’s suggestions, we have mentioned the challenges of sustainable practices in the second paragraph of Introduction (Line 50-62) and Literature review (Line 138-144).
Point 14: L. 462. How do we know that official websites are valid, rather than advertising?
Response 14: Based on reviewer’s suggestions, we have modified the Method part and detailed why we used website content analysis, “restaurants usually operate their official websites as a marketing tool for providing rich information to customers. This study thus conducts a website content analysis .... This method is appropriate for compiling a complete list of attributes for evaluation and for studying the online content of destination restaurants, as this sector has a substantial online presence” (Line 263-268). It is true that some sustainable practices mentioned on the websites while not be implement, and some practices are conducted but might be not indicated online. Therefore, we indicate it as a limitation of the study, “This study has limitations in only analysing the sustainability of M3SRs by official website content but not researching other types of fine-dining restaurants and other information on sustainable practices. Some restaurants do not renew their website to date, and some even do not have official website. Therefore, some practices are missed due to the limited information on their official websites. On the other hand, whether the sustainable practices that are informed on websites are conducted in long term or not remains unknown.” (Line 599-605).
Point 15: Overall, the paper attempts to use Michelin restaurants as a way of looking at sustainability. Unfortunately the paper depends on what is presented on restaurant websites – we have no idea whether that information is valid. The overall presentation should include both positive and negative information – for example, why do so few restaurants practice certain sustainable practices? In its current form the paper sounds like more advertising for the Michelin restaurants. And the authors need to acknowledge that many customers looking for good food are turned off by the emphasis on prestige and the practice of very high prices. Many customers prefer the authentic local restaurant run by the same people for 30 years. This paper needs a very major revision.
Response 15: We appreciate the reviewer’s helpful comments. We have revised the article accordingly and marked the revision in red.
Reviewer 3 Report
The paper is very interesting. The research design is linear.
I would recommend improving clarify to explicit the research questions: For better clarity, it would be useful to explain the research questions possibly at the end of the introduction.
The literature is thorough and up-to-date.
In the section method, there is a need to implement adequate literature related to website evaluation criteria: In my opinion, in the description of the method it would be necessary to identify web site analysis tools recognized by the literature to give greater scientific solidity to your research with a description of the criteria.
The results must be implement describing in depth the various items investigated, for example on the different adoption of sustainable practices by geographical area: Another section that should be implemented concerns results. Table 2 provides a summary picture of the sustainability practices of starred restaurants, but it would be useful to enrich this section with other elaborations possibly made during the research, trying to link the results with those discussed in the subsequent discussion. Furthermore, it would be interesting to analyze the correlation between sustainability practices and the geographical area of restaurants.
The discussion is well structured: The discussion was well structured in the six sub-paragraphs.
Author Response
We would like to express our gratitude to the editors and the reviewers for their helpful comments and suggestions for improving the quality of our paper. We have revised and rewritten our paper accordingly. All major changes have been highlighted in red in the document.
Point 1: The paper is very interesting. The research design is linear.
Response 1: We appreciate the reviewer’s positive comment.
Point 2: I would recommend improving clarify to explicit the research questions.
Response 2: Based on the reviewer’s suggestion, we have clarified the research questions in the Introduction: “1) the extent to which sustainable practices are promoted as part of MSRs’ offerings; and 2) how these practices reflect the locality” (Line 100-102).
Point 3: In the section method, there is a need to implement adequate literature related to website evaluation criteria.
Response 3: According to the reviewer’s suggestions, we have revised Method section by explaining why website content analysis is applied in the first paragraph of Method, “restaurants usually operate their official websites as a marketing tool for providing rich information to customers. This study thus conducts a website content analysis .... This method is appropriate for compiling a complete list of attributes for evaluation and for studying the online content of destination restaurants, as this sector has a substantial online presence” (Line 263-268). We have also detailed the criteria definition and coding of identified sustainable practices at the end of Method section, “The website contents of each M3SRs are analysed to determine how sustainability restaurants put forward to attract customers and how the identified sustainable practices are communicated online. The criteria for determining whether a restaurant is representative of each element is defined and coded as noted in Table 2.” (Line 285-291).
Point 4: The results must be implement describing in depth the various items investigated, for example on the different adoption of sustainable practices by geographical area.
Response 4: Based on the reviewer’s suggestions, we have merged the Results and Discussion section and provided in depth discussion. We have explained why some percentages are low and tied up findings with literature. For example, from Line 294 to 301, we have explained why all sustainable initiatives are mentioned by less than half M3SRs, “Some possible explanations would be that 1) promoting sustainability might be incompatible with providing fine-dining experience because importing off-season foodstuffs could not be sustainable; 2) some websites are just used as online reservation platforms with limited information to introduce their sustainable initiatives” From line 335-341, we have mentioned that local food has not been incorporated with mainstream fine-dining restaurants because it seems not enough to display the exclusiveness and the premium price of fine dining. Similar explanations in other sustainable practices are also marked in red, including farm-to-table activity (Line 344-346), food foraging (Line 387-394), efficient restaurant (Line 412-416), sustainable menu (Line 418-421). We have also clarified our findings that could support previous studies on minimising waste, like food waste are generated before consumption in fine-dining restaurants (Line 444-446) and disposables in fine-dining restaurants are less studied while the use of disposables has been increased (Line 452-459). More detailed discussions are also provided for restaurant history (Line 463-468), and cooking school (Line 533-540).
As for the different adoption of sustainable practices by geographical area, we suppose the sample of Micheline three-star restaurants cannot reflect the geographical differences because the European restaurants are much more than others. It is one of the limitations of the study, “the awarded M3SRs are all fine-dining restaurants evaluated by the criteria of the Michelin Restaurant Guide, and most of them are in European countries due to the Michlin Guide’s originality” (Line 605-607).
Point 5: The discussion is well structured.
Response 5: We appreciate the reviewer’s positive comment.
Round 2
Reviewer 1 Report
Thank you for addressing my raised concerns. Good work.
Author Response
We appreciate the reviewer’s positive comment.
Reviewer 2 Report
The authors have made some changes to their original manuscript. But the largest part of the presentation of Results is examples of the small percentages of restaurants which practice some sustainability behavior. This misrepresents that most of these restaurants do not practice the sustainability behaviors investigated. The focus of the article should be on why these Michelin starred restaurants do NOT practice sustainabilty behaviors. Since we are not sure on the validity of these restaurant websites, I do not see the value of these data. Further, the authors continue to focus on the sustainability behaviors, even when most Michelin star restaurants do not practice them. If these restaurants are as influential as the authors claim, then their failure to embrace sustainability (as leaders) deserves study and discussion
Author Response
According to the reviewer’s comments, we have further provided M3SRs’ failure to embrace sustainability with three possible explanations; “This phenomenon has some possible explanations. First, some websites are used as online reservation platforms with limited information to introduce the restaurant philosophy or practices, including sustainability, such as Zén in Singapore, Sushi Yoshitake in Japan, and Benu in the USA. Second, providing luxury dining experience appears to be incompatible with sustainability, given that many fine-dining restaurants keep searching luxury and off-season ingredients from distant regions and produce much more food waste than other types of restaurants just due to aesthetics concerns. Third, this study argues that the current model of fine-dining restaurants seems to go the opposite way that far from sustainability. Food critics have begun to question the sustainability of fine-dining restaurants and their performance of a rococo act for a rarefied audience that ‘forever trying to dazzle self-regarding epicures with new stunts, novel sensations, modes of presentation we hadn’t imagined, flora and fauna rarely pinned down on a plate’. Some recent news has also reinforced the criticism, including Noma in Copenhagen announced that they will close the current restaurant in 2025 and ‘create a lasting organization dedicated to ground-breaking work in food’, Amass in Copenhagen was taken under bankruptcy, which is a true lead-er in sustainability with Michelin Green Star, as well as the release of a satirical film, The Menu, which interrogates the culture created by fine-dining restaurants and chefs” (Line 302-319). Additionally, we think the sustainable practices presented on the rest of M3SRs’ websites are still worth investigating because “they seem to demonstrate sustainability online in an acceptable way by the Michelin Guide judges, food critics, foodies, and the restaurant industry and act as sustain-ability ambassador in the food system by food media and cookbooks” (Line 320-323). We hope the revision could address the reviewer’s concern. We have revised the article accordingly and marked the revision in red.
Reviewer 3 Report
The new version of this article has improved. Each section has been implemented according to some recommendations given by the reviewers. In my opinion, this paper can be published
Author Response

(The authors gave the same response as above.)

Round 3
Reviewer 2 Report
The authors have added a good section at the beginning of Results. Other than that, the paper is basically unchanged, and my criticism of Method and Results still holds.
Author Response
We appreciate the reviewer’s previous comments. However, we do not fully agree with the reviewer’s concerns about the methods and results for three reasons. First, the research questions are reasonably proposed and addressed. This study aims to address two research questions: “1) the extent to which sustainable practices are promoted as part of MSRs’ offerings; and 2) how these practices reflect the locality” (Line100-102). For the first question, the results reflect that “sustainability has not been considered a necessary offering in M3SRs” (Line 300), and possible explanations are discussed. In addition, it is important to highlight that we explicitly focus on the extent, e.g., what is the degree to which such restaurants promote sustainability practices online. For the second question, we argue that the identified practices could indicate how fine-dining restaurants promote locality in sustainable practices with evidence. Second, this study also focuses on discussing how Michelin-starred restaurants self-present sustainability by promoting locality. Previous studies have proposed various measurements to evaluate restaurant’s sustainability, “such as restaurant and food service life cycle assessment, and Green Restaurants ASSessment (GRASS)” (Line 44-46), while this study extracts the sustainable practices that related to locality because “localism has been advocated in the restaurant industry” (Line 75-76) and “ locality has become an increasingly prevalent food trend in recent years which could be regarded as a form of local cultural capital that strengthens sense of place, place and product differentiation, and place branding in a globalised world” (Line 86-88). Therefore, even though “identified sustainable practices are mentioned by less than 50% of M3SRs” (Line 294), many M3SRs “promotes locality to bridge sustainability and luxury dining experience” (Line 328). They are worthy of investigating “because they seem to demonstrate sustainability online in an acceptable way by the Michelin Guide judges, food critics, foodies, and the restaurant industry and act as pioneers in leading food trends in the food system by food media and cookbooks” (Line 323-326). Third, to address the second research question, this study examining how these M3SRs’ sustainable practices promotes locality could “potentially provide insights into the compatibility between luxury dining experience and sustainability as well as the possible influence of M3SRs’ as culinary leaders on the food-service industry and other types of restaurants in practising sustainability” (Line 328-331). Overall, we argue that even though less than half of identified M3SRs present identified sustainable practices on their websites, this study could still provide insights into the food trends of promoting locality to improve the sustainability of fine-dining restaurants. In addition, we would note that website content analysis is a widely used research method while we also note with respect to future research interviews, surveys and field inspection/observation are appropriate methods by which to compare what is promoted on the restaurant websites compared to what actually happens at the restaurant (Line 631-633).